

# Bacterial diversity and community in Qula from the Qinghai–Tibetan Plateau in China

Yan Zhu[1,*], Yingying Cao[1,*], Min Yang[2], Pengchen Wen[1], Lei Cao[1], Jiang Ma[1], Zhongmin Zhang[1] and Weibing Zhang[1]

[1] College of Food Science and Engineering, Gansu Agricultural University, Lanzhou, Gansu, China
[2] College of Science, Gansu Agricultural University, Lanzhou, Gansu, China
* These authors contributed equally to this work.

## ABSTRACT

Qula is a cheese-like product usually prepared with unpasteurized yak milk under open conditions, with both endogenous and exogenous microorganisms involved in the fermentation process. In the present study, 15 Qula samples were collected from five different regions in China to investigate the diversity of microbial communities using high-throughput sequencing targeting the V3–V4 region of 16S rRNA gene. The bacterial diversity significantly differed among samples of different origins, indicating a possible effect of geography. The result also showed that microbial communities significantly differed in samples of different origin and these differences were greater at the genus than the phylum level. A total of six phyla were identified in the samples, and Firmicutes and Proteobacteria had a relative abundance >20%. A total of 73 bacterial genera were identified in the samples. Two dominant genera (*Lactobacillus* and *Acetobacter*) were common to all samples, and a total of 47 operational taxonomic units at different levels significantly differed between samples of different origin. The predicted functional genes of the bacteria present in samples also indicated differences in bacterial communities between the samples of different origin. The network analysis showed that microbial interactions between bacterial communities in Qula were very complex. This study lays a foundation for further investigations into its food ecology.

Corresponding author
Weibing Zhang,
zzz888666@gsau.edu.cn

## INTRODUCTION

The yak is the only bovine species adapted to the cold and harsh conditions of the Hindu Kush-Himalayan region and the Qinghai–Tibetan Plateau with its altitude range of 2,000–5,000 m above sea level (*Cui et al., 2016*). In China, there are about 14 million yak distributed in Yunnan, Xizang, Qinghai, Gansu, Sichuan, and other provinces, accounting for 95% of the world's yak population. Yak have been central to the development of the farming and pastoral communities of these areas (*He et al., 2011*).

Yak produce milk with high nutritive value, which is considered a high-quality raw material for manufacturing food such as yak butter, milk powder, cheese, and fermented milk (*He et al., 2011*). Qula is a traditional homemade fermented cheese-like yak milk

product, which has been made and consumed in Yunnan, Xizang, Qinghai, Gansu, and Sichuan for thousands of years (*Li et al., 2010*; *He et al., 2011*). Due to the dispersion of yak milk, fresh yak milk is often processed to Qula on local yak farms, and can be stored for years (*Li et al., 2010*). Qula is white or yellow grain, and contains 7–12% moisture and approximately 80% protein. Nowadays, Qula is usually collected and transported to a factory to make yak milk casein. Because Qula is usually prepared with unpasteurized yak milk under open conditions, both endogenous and exogenous microorganisms are involved in the fermentation process. In the past several years, microbial species in Qula has been traditionally identified by conventional culture and molecular biology methods (*Duan et al., 2008*; *Zhang et al., 2015*). However, it is obvious that these methodologies are not optimal to provide detailed information regarding the microbial communities in complex matrixes for technique limitations (*Li et al., 2011*; *Ao et al., 2012*; *Ding et al., 2017*).

High-throughput sequencing technologies such as Illumina MiSeq sequencing are now powerful tools for better assessment of microbial diversity due to low cost and high read quantity and quality (*Liu et al., 2007*; *Caporaso et al., 2012*; *Kryachko et al., 2012*; *Solieri, Dakal & Giudici, 2013*; *Lenchi et al., 2013*; *Tago et al., 2014*). Numerous studies have been carried out to determine microbial communities of fermented food including Plaisentif cheese, salami, Pu-erh tea, and douchi (*Polka et al., 2015*; *Dalmasso et al., 2016*; *Zhang et al., 2016*; *Yang et al., 2016*; *Zhang et al., 2018*). However, relatively little research has addressed the microbial community structure in traditional Qula using modern culture-independent molecular techniques, especially Illumina MiSeq sequencing. This study's objective was to make an inventory of the diversity of microbial communities in Qula from different regions in China using Illumina MiSeq approaches. The data generated, particularly the differences in the distribution of particular taxonomic groups, were used to evaluate the effect of the processing environments on the bacterial communities.

## MATERIALS AND METHODS

### Sample collection

Qula samples were collected in October 2016. Homemade Qula samples were purchased directly from different sellers at local markets in five regions of China, samples #1-3 are not representative of triplicate samples from a single Qula (Table 1). All samples were placed into sterile tubes, numbered, and placed in an ice box for transportation to the laboratory for extraction of DNA.

### DNA extraction

DNA was extracted from 0.2 g of the Qula samples using an E.Z.N.A. Soil DNA Kit D5625-01 (OMEGA, Norcross, GA, USA) according to the manufacturer's instructions. The extracted DNA was quantified using a Qubit 2.0 spectrophotometer (Invitrogen, Carlsbad, CA, USA), and the integrity of the extracted DNA from the Qula samples was confirmed by electrophoresis in a 0.8% agarose gel.

**Table 1 Information of Qula samples from different origins.**

| Sample ID | Color | Sample location | Origin |
|---|---|---|---|
| YN1 | White | Yunnan province, China | Xiangelila (27.31N 95.52E) |
| YN2 | White | Yunnan province, China | Xiangelila (27.31N 95.52E) |
| YN3 | White | Yunnan province, China | Xiangelila (27.31N 95.52E) |
| XZ1 | Yellow | Xizang province, China | Shannan (29.23N 91.77E) |
| XZ2 | Yellow | Xizang province, China | Shannan (29.23N 91.77E) |
| XZ3 | Yellow | Xizang province, China | Shannan (29.23N 91.77E) |
| QH1 | Yellow | Qinghai province, China | Huangzhong (36.51N 101.57E) |
| QH2 | Yellow | Qinghai province, China | Huangzhong (36.51N 101.57E) |
| QH3 | Yellow | Qinghai province, China | Huangzhong (36.51N 101.57E) |
| GN1 | Yellow | Gansu province, China | Gannan (34.98N 102.92E) |
| GN2 | Yellow | Gansu province, China | Gannan (34.98N 102.92E) |
| GN3 | Yellow | Gansu province, China | Gannan (34.98N 102.92E) |
| SC1 | Yellow | Sichuan province, China | Kangdin (30.05N 101.97E) |
| SC2 | Yellow | Sichuan province, China | Kangdin (30.05N 101.97E) |
| SC3 | Yellow | Sichuan province, China | Kangdin (30.05N 101.97E) |

## Illumina MiSeq sequencing

Next generation sequencing library preparations and Illumina MiSeq sequencing was conducted at Shanghai Personal Biotechnology Co. Ltd. (Shanghai, China). The primers 338F (ACTCCTACGGGAGGCAGCA) and 806R (GGACTACHVGGGTWTCTAAT) with barcodes were employed to amplify the bacterial 16S rRNA genes from the extracted DNA (*Feng et al., 2017*). The PCR amplification was conducted using high-fidelity Trash Start Fastpfu DNA Polymerase (TransGen Biotech, Beijing, China). The thermal cycle conditions were as follows: 5 min at 95 °C; followed by 25 cycles of 30 s at 95 °C, 30 s at 56 °C, and 40 s at 72 °C; then 10 min at 72 °C, after which the samples were cooled at 4 °C.

## Processing of high-throughput sequencing data

Amplicons were sequenced using a paired-end method by Illumina MiSeq with a six cycle index read. Raw data generated from the high-throughput sequencing run were processed and analyzed following the pipelines of Mothur (V.1.31.2) and QIIME (V1.7.0) (*Schlosset et al., 2009*; *Caporaso et al., 2012*). Sequence reads were trimmed so that the average Phred quality score for each read was above 20. After trimming, these reads were assembled using the Flash software (V.1.2.7) (*Fu et al., 2015*) and reads that could not be assembled were discarded. Chimera sequences were identified and removed using UCHIME (V.4.2) (*Edgar et al., 2011*). Quality sequences were clustered into operational taxonomic units (OTUs) using a 97% identity threshold with QIIME's uclust (*Fu et al., 2015*).

The taxonomic identities of bacterial OTU representative sequences were classified with the Ribosomal Database Project classifier with the SILVA databases (*Wang et al., 2007*). This dataset is available in the SRA at the NCBI under accession number SRP128600.

## Diversity and statistical analysis

The relative abundance (%) of individual taxa within each community was estimated by comparing the number of sequences assigned to a specific taxon to the number of total sequences obtained for that sample. Alpha diversity analysis, which included the Simpson, Chao1, and Shannon indices, was performed using the summary single command of the MOTHUR software (V.1.31.2, http://www.mothur.org/). One-way ANOVA followed by Bonferroni's multiple comparison post hoc tests was used to explore variations in Chao1 richness index and Shannon index.

Unweighted UniFrac distances principal coordinates analysis (unweighted UniFrac PCoA) was performed using the R software package (v 2.15.3; https://www.r-project.org/) on the basis of the relative abundance of bacterial genera. Unweighted pair-group method with arithmetic means clustering was also performed using QIIME 1.7.0 (*Caporaso et al., 2010*) and displayed using R software (v 2.15.3; https://www.r-project.org/). The Adonis permutational multivariate analysis (Adonis/PERMANOVA) and analysis of similarities (ANOSIM) were used to assess the differences of bacterial community structure in samples.

Differentially abundant features among the different samples were identified using the linear discriminant analysis (LDA) effect size (LEfSe) pipeline (http://huttenhower.sph.harvard.edu/galaxy/) on the OTU level (relative abundance >1%). Taxa with significant differential abundances were detected by the non-parametric factorial Kruskal–Wallis (KW) rank sum test. The LEfSe analysis was performed with an alpha value for the KW test of 0.05 and a threshold on the logarithmic LDA score of 2.0 (*Xu et al., 2017*).

Phylogenetic investigation of communities by reconstruction of unobserved states software (PICRUSt, v1.0, http://picrust.github.io/picrust/) was used to predict metagenome functional features from 16S rRNA genes (*Langille et al., 2013*; *Chen et al., 2015*; *Wu et al., 2016*). The threshold of Picking OTUs for use in PICRUSt is 97%. Predicted functional pathways were annotated by using the Kyoto Encyclopedia of Genes and Genomes (KEGG) (*Kanehisa et al., 2012*; *Langille et al., 2013*). OTUs for which a KEGG profile could not be retrieved constitute the fraction of unexplained taxonomic units (FTU). The abundance of the retrieved KEGG metabolic profiles in samples from different regions were reported as mean ± standard error of the mean. One-way ANOVA followed by Duncan' test was used to determine significant differences between the samples and differences were considered significant when $P < 0.05$.

The top 50 bacterial genera in Qula samples were used for network analysis. Spearman's rank correlations between selected genera were calculated using the R package (*Ihaka & Gentleman, 1996*). A valid co-occurrence was selected as a strong correlation if the Spearman's correlation coefficient ($\rho$) was greater than 0.6 with a significance level less than 0.01 (*Barberán et al., 2012*; *Huang et al., 2017*). Correlation networks between genera in the samples were constructed and visualized in the Cytoscape software (v 3.2.1; http://www.cytoscape.org/) (*Shannon et al., 2003*).

**Table 2 OTUs, Good's Coverage, Chao1, and Shannon's indices for 16S rRNA sequencing of the samples.**

| Sample ID | Reads | OTU | Good's Coverage | Chao1 | Shannon |
|---|---|---|---|---|---|
| YN1 | 26688 | 47 | 97.88% | 194 | 2.32 |
| YN2 | 25003 | 55 | 97.93% | 181 | 2.07 |
| YN3 | 25344 | 33 | 96.92% | 184 | 2.16 |
| XZ1 | 23110 | 116 | 97.36% | 395 | 3.68 |
| XZ2 | 23907 | 114 | 98.56% | 409 | 3.73 |
| XZ3 | 24756 | 133 | 98.26% | 437 | 3.86 |
| QH1 | 24869 | 142 | 97.45% | 305 | 4.12 |
| QH2 | 21213 | 179 | 96.59% | 268 | 4.23 |
| QH3 | 23102 | 175 | 97.84% | 297 | 4.48 |
| GN1 | 26439 | 101 | 97.34% | 223 | 2.43 |
| GN2 | 24801 | 98 | 98.48% | 260 | 2.73 |
| GN3 | 24688 | 110 | 97.76% | 252 | 2.61 |
| SC1 | 22156 | 79 | 96.45% | 421 | 5.04 |
| SC2 | 21424 | 93 | 97.38% | 439 | 5.08 |
| SC3 | 20087 | 101 | 97.58% | 391 | 4.91 |

# RESULTS

## Sequencing and classification

Total DNA was extracted from the Qula samples, after which V3–V4 of the 16S rRNA gene was PCR amplified from each DNA sample. PCR products were sequenced using the paired-end method by Illumina MiSeq. After quality control, a total of 302,930 high-quality 16S rRNA gene sequences were recovered from the samples (Table 2). The average lengths of high-quality sequences were 455 bp for the bacterial community. The high-quality sequences were grouped into 1,576 OTUs at the 97% similarity level and, after removing singletons, the average number of OTUs was 105. The samples from Qinghai had the highest diversity of bacterial communities (165 OTUs), and samples from Yunnan had the lowest (45 OTUs).

In the study, the coverage of all samples was in the range of 96.45–98.56%, indicating an adequate level of sequencing to identify most diversity in the samples. The rarefaction curve for Shannon diversity indices was nearly parallel with the $x$-axis (Fig. 1) and the Shannon diversity index reached saturation, suggesting that although new phylotypes would be expected with additional sequencing, the majority of bacterial phylotypes present in Qula had been captured.

## Analysis of alpha diversity

Two indices were determined (Chao1 richness index and Shannon index) to measure the alpha diversity of the microbiome in the analyzed sample (Table 2). The microbial diversity was compared, as estimated by the Shannon index, and the result showed that the bacterial diversity difference was statistically significant ($P < 0.001$, Bonferroni's test) in Qula samples from different origins, indicating that geography may have impacted

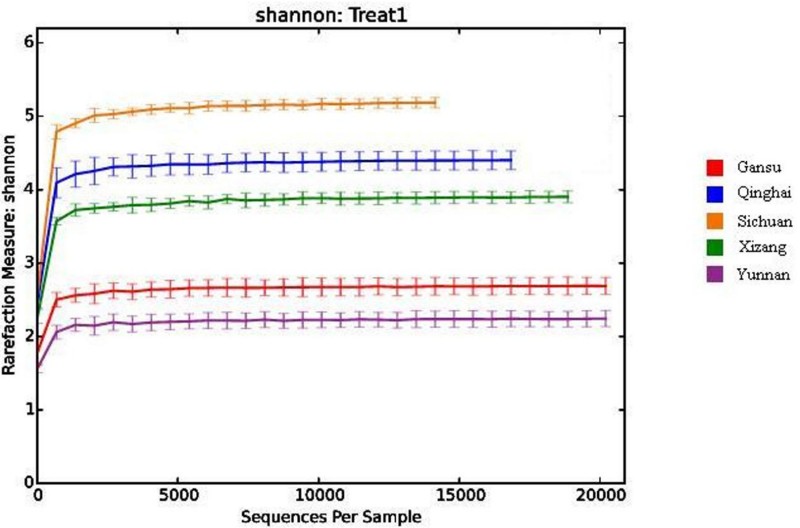

**Figure 1** Rarefaction curves for Shannon diversity indices of bacteria in the samples.

the bacterial diversity in the samples. The samples from Sichuan had the highest bacterial diversity (Table 2), implying more abundant bacteria compared with samples from other regions. We also found that bacterial diversity was lowest in samples from Yunnan (Table 2).

We also compared the bacterial species richness, as estimated by the Chao1 richness index (Table 2). The samples from Sichuan had the highest bacterial species richness, while the bacterial species richness of the samples from Yunnan is the lowest. The bacterial species richness was significantly different ($P < 0.001$, Bonferroni's test) in samples from different origins except for Qula samples collected in Xizang and Sichuan ($P = 1.000$, Bonferroni's test).

## Comparison of bacterial communities in samples

We examined the compositions of bacterial communities in the samples. A total of six phyla were identified in the samples, and Firmicutes and Proteobacteria had a relative abundance >20% (Table 3; Fig. S1). The relative abundance of Firmicutes, the most abundant phylum in all samples, was in the range of 45.7–79.2%. Proteobacteria was the second most abundant phylum, with a relative abundance of 20.80–54.18%. Actinobacteria was the third most abundant phylum in samples from Xizang and Qinghai, but its relative abundance was below 1% in other samples. The relative abundance of the other three phyla was lower than 1% and differed between samples from different regions.

A total of 73 bacterial genera were identified in the samples. The abundance of the main bacterial genera in the samples is shown in Table 3 and Fig. S2. A total of 10 dominant genera were observed in all samples, all with a relative abundance exceeding 1%. In the samples from Yunnan, *Lactobacillus* and *Acetobacter* were the dominant genera, with relative abundance of 79.12 and 17.92%, respectively. Five dominant genera (*Acetobacter*, *Lactococcus*, *Lactobacillus*, *Leuconostoc*, and *Enterococcus*) were observed in

**Table 3 Percentage of the main bacterial phylum and genus in Qula samples from different regions in China.**

| Bacteria | Percentage composition in samples | | | | |
|---|---|---|---|---|---|
| | Yunnan $n = 3$ | Xizang $n = 3$ | Qinghai $n = 3$ | Gansu $n = 3$ | Sichuan $n = 3$ |
| **Phylum** | | | | | |
| Firmicutes | 79.20 | 51.87 | 64.70 | 45.65 | 57.64 |
| Proteobacteria | 20.80 | 45.91 | 32.7 | 54.18 | 42.24 |
| Actinobacteria | 0.00 | 2.13 | 1.62 | 0.09 | 0.08 |
| Acidobacteria | 0.00 | 0.03 | 0.66 | 0.03 | 0.05 |
| Bacteroidetes | 0.00 | 0.05 | 0.23 | 0.01 | 0.00 |
| Cyanobacteria | 0.00 | 0.01 | 0.04 | 0.03 | 0.00 |
| Unidentified | 0.00 | 0.01 | 0.00 | 0.00 | 0.00 |
| **Genus** | | | | | |
| *Lactobacillus* | 79.12 | 5.41 | 14.48 | 36.81 | 13.15 |
| *Acetobacter* | 17.92 | 41.43 | 14.06 | 50.66 | 5.84 |
| *Lactococcus* | 0.00 | 41.09 | 23.37 | 7.71 | 26.63 |
| *Leuconostoc* | 0.00 | 3.52 | 18.97 | 0.10 | 15.26 |
| *Streptococcus* | 0.00 | 0.01 | 5.43 | 0.01 | 1.25 |
| *Acinetobacter* | 0.00 | 0.08 | 4.58 | 0.45 | 2.09 |
| *Burkholderia* | 0.01 | 0.12 | 3.83 | 0.16 | 0.10 |
| *Ralstonia* | 0.00 | 0.04 | 1.32 | 0.05 | 0.03 |
| *Rhodanobacter* | 0.00 | 0.03 | 1.20 | 0.02 | 0.03 |
| *Pseudomonas* | 0.01 | 0.32 | 0.61 | 0.00 | 23.83 |
| *Enterococcus* | 0.00 | 1.13 | 0.46 | 0.15 | 0.01 |
| *Geobacillus* | 0.00 | 0.01 | 0.39 | 0.00 | 0.00 |
| *Sphingomonas* | 0.00 | 0.01 | 0.37 | 0.01 | 0.01 |
| *Anoxybacillus* | 0.00 | 0.00 | 0.33 | 0.02 | 0.00 |
| *Bacillus* | 0.05 | 0.14 | 0.30 | 0.02 | 0.00 |
| *Rhodococcus* | 0.00 | 0.02 | 0.27 | 0.02 | 0.01 |
| *Gluconobacter* | 0.81 | 0.63 | 0.24 | 0.93 | 0.90 |
| *Agrobacterium* | 0.00 | 0.01 | 0.23 | 0.00 | 0.00 |
| *Ochrobactrum* | 0.00 | 0.01 | 0.19 | 0.01 | 0.01 |
| *Comamonas* | 0.00 | 0.00 | 0.17 | 0.00 | 0.00 |
| Others | 2.07 | 5.99 | 9.20 | 2.86 | 10.85 |

samples from Xizang, with relative abundance in the range of 1.13–41.43%. Among them, *Acetobacter* and *Lactococcus* were the first and second most abundant genera, representing 82.52% of the bacterial population. Nine genera (*Lactococcus*, *Leuconostoc*, *Lactobacillus*, *Acetobacter*, *Streptococcus*, *Acinetobacter*, *Burkholderia*, *Ralstonia*, and *Rhodanobacter*) were dominant in samples from Qinghai, with relative abundance of 23.37%, 18.97%, 14.48%, 14.06%, 5.43%, 4.58%, 3.83%, 1.32%, and 1.20%, respectively. In samples from Gansu, there were three dominant genera (*Acetobacter*, *Lactobacillus*, and *Lactococcus*) with 50.66%, 36.81%, and 7.71%, respectively. Seven bacterial genera

## Cladogram

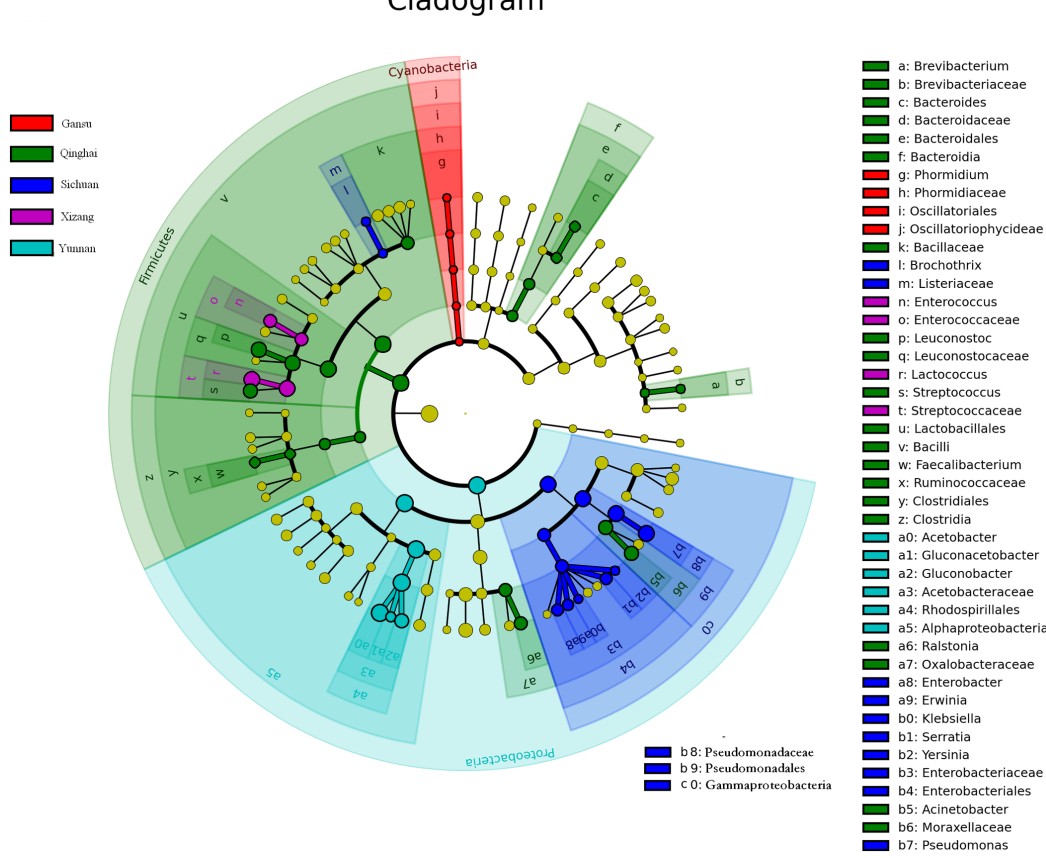

**Figure 2 Linear discriminant analysis of microbial community compositions in Qula samples.** The node size represents the difference in relative abundance. Yellow nodes indicate OTUs with no significant differences in relative abundance. Red, green, blue, purple, and blue–green nodes indicate OTUs with significant differences in Qula samples from Gansu, Qinghai, Sichuan, Xizang, Yunnan. The meaning of shading color is the same as node color.

(*Lactococcus, Pseudomonas, Leuconostoc, Lactobacillus, Acetobacter, Acinetobacter,* and *Streptococcus*) were dominant in samples from Sichuan, with relative abundance 26.63%, 23.83%, 15.26%, 13.15%, 5.84%, 2.09%, and 1.25%, respectively. Additionally, a total of 63 non-dominant bacterial genera were present in the samples, all with a relative abundance less than 1%.

 Linear discriminant analysis showed a significant difference in bacterial community compositions in Qula from different regions (Fig. 2). We observed that a total of 47 OTUs (six in samples from Yunnan, 20 from Qinghai, four from Xizang, four from Gansu, and 13 from Sichuan) significantly differed between samples from different origins. Of Firmicutes, six bacterial genera significantly differed between samples; and11 genera of Proteobacteria differed.

### Analysis of beta diversity

The beta diversity results of unweighted UniFrac PCoA (Fig. 3) based on the relative abundance of bacterial community showed that most samples from the same location were closely grouped together except for QH1. The sample (QH1) cluster with the
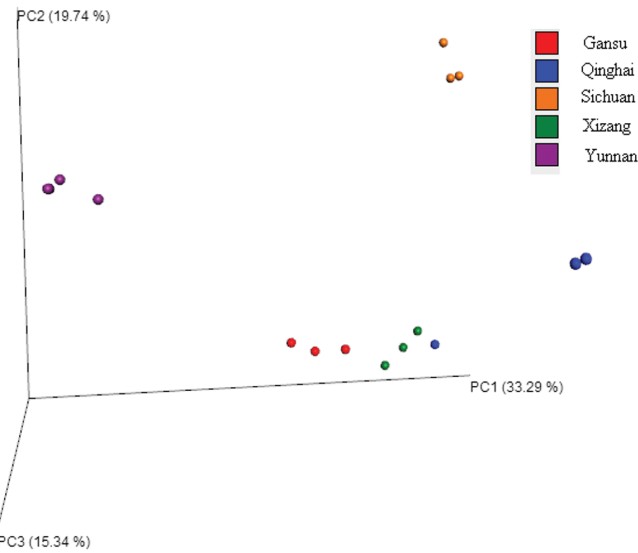

**Figure 3 Principal coordinate analysis (PCoA) of bacteria genera based on the unweighted UniFrac distance among samples.**

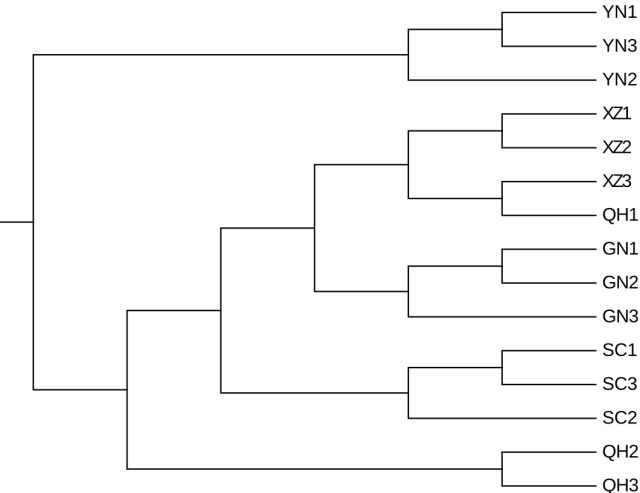

**Figure 4 Unweighted pair-group analysis (UPGMA) of bacteria using arithmetic means based on unweighted UniFrac analysis.**

XZ samples, indicating that the microbiota in this sample (QH1) may be affected by other factors such as breed, diet, or parity. The results were much similar when comparing samples from different regions using unweighted pair-group analysis (Fig. 4). The Adonis/PERMANOVA was scored with $P = 0.001$ ($P < 0.05$, in significance level). The results of ANOSIM's statistical analysis showed that the production region was a salient factor affecting the bacterial composition of the different samples ($R = 0.9304$, $P = 0.001$). A heatmap analysis (Fig. 5) of the top 50 bacterial genera also indicated a significant difference in bacterial communities between samples.

Thus, we concluded that the bacterial flora in the samples was stratified by geographic region. The Qula samples used in this study were produced by similar traditional methods.

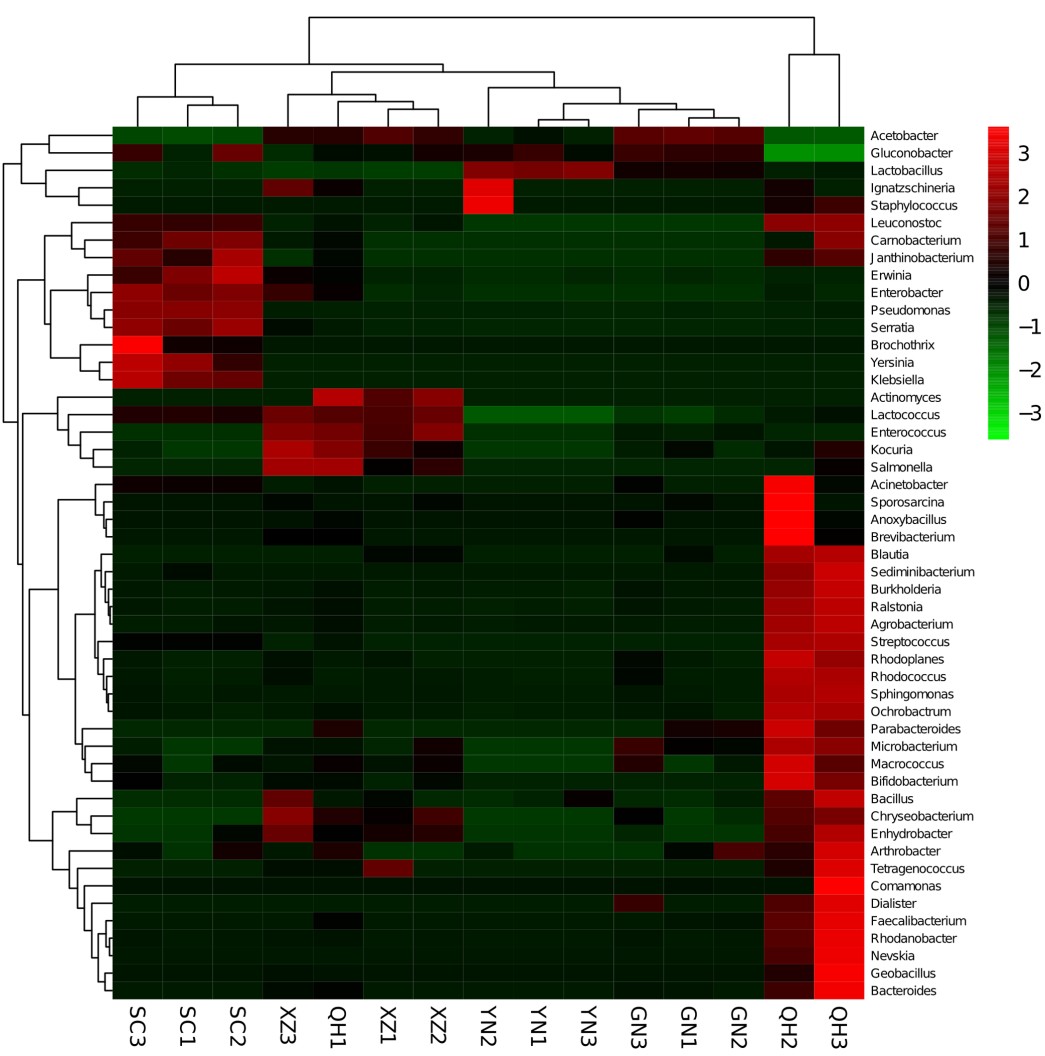

**Figure 5 Heatmap and dendrogram of top 50 bacterial genera present in the samples.** Color blocks represent the relative abundance of genera, namely the Z-value. More red indicates a higher relative abundance.

Therefore, it is likely that the differences in microbial structure between samples of different origin were due to geographic location and other environmental factors.

## Functional genes of the bacteria present and their metabolism in Qula samples

The PICRUSt software was used to predict the functional genes of the bacteria present in samples and their metabolic pathways. The FTU is shown in Table S1. The fraction was highest in the samples from Sichuan (14.35 ± 1.64%) and lowest in the samples from Yunnan (3.46 ± 0.29%). For the samples from other origins, the FTU was not significant ($P > 0.05$, Duncan' test).

Of the functional microbial genes in the samples, 44.17–50.37% were related to the following metabolic pathways: amino acid (AA) metabolism, carbohydrate metabolism, energy metabolism, metabolism of cofactors and vitamins, nucleotide metabolism,

**Table 4 Functional features relating to metabolism of genes from bacteria in the samples.**

| Metabolic pathway | Percentage composition in samples | | | | |
| --- | --- | --- | --- | --- | --- |
| | Xizang<br>$n = 3$ | Sichuan<br>$n = 3$ | Qinghai<br>$n = 3$ | Gansu<br>$n = 3$ | Yunnan<br>$n = 3$ |
| Amino Acid Metabolism | 10.00 ± 0.08[a] | 9.75 ± 0.08[b,c] | 9.83 ± 0.12[a,b] | 9.49 ± 0.04[d] | 6.81 ± 0.2[e] |
| Biosynthesis of Other Secondary Metabolites | 0.87 ± 0.01[a] | 0.70 ± 0.01[d] | 0.83 ± 0.03[b] | 0.82 ± 0.00[b,c] | 0.50 ± 0.02[e] |
| Carbohydrate Metabolism | 10.77 ± 0.04[a,b] | 9.60 ± 0.03[e] | 10.66 ± 0.13[c,d] | 10.87 ± 0.02[a] | 10.72 ± 0.01[b,c] |
| Energy metabolism | 5.30 ± 0.11[b,c] | 4.75 ± 0.01[e] | 5.08 ± 0.15[d] | 5.87 ± 0.04[a] | 5.35 ± 0.04[b] |
| Enzyme families | 1.82 ± 0.01[e] | 1.89 ± 0.01[c,d] | 1.91 ± 0.07[c] | 2.01 ± 0.01[b] | 2.35 ± 0.03[a] |
| Glycan biosynthesis and metabolism | 1.96 ± 0.01[a,b,c] | 1.98 ± 0.00[a] | 1.93 ± 0.03[d] | 1.89 ± 0.00[e] | 1.97 ± 0.00[a,b] |
| Lipid metabolism | 3.19 ± 0.02[c,d] | 3.43 ± 0.02[a] | 3.41 ± 0.02[a,b] | 3.21 ± 0.03[c] | 2.61 ± 0.05[e] |
| Metabolism of cofactors and vitamins | 4.50 ± 0.12[a] | 3.82 ± 0.03[c,d] | 4.00 ± 0.35[c] | 4.48 ± 0.02[a,b] | 3.39 ± 0.09[e] |
| Metabolism of other amino acids | 1.97 ± 0.02[a,b,c] | 1.95 ± 0.01[b,c,d] | 1.99 ± 0.03[a,b] | 2.00 ± 0.01[a] | 1.60 ± 0.03[e] |
| Metabolism of terpenoids and polyketides | 1.89 ± 0.00[a,b,c] | 1.93 ± 0.00[a,b] | 1.96 ± 0.08[a] | 1.84 ± 0.01[c,d] | 1.72 ± 0.01[e] |
| Nucleotide metabolism | 4.23 ± 0.05[b,c] | 3.65 ± 0.03[e] | 4.05 ± 0.16[d] | 4.24 ± 0.03[b] | 4.74 ± 0.04[a] |
| Xenobiotics biodegradation and metabolism | 3.54 ± 0.09[a,b] | 3.30 ± 0.03[d] | 3.47 ± 0.14[a,b,c] | 3.57 ± 0.03[a] | 2.59 ± 0.07[e] |

**Note:**
Letters [(a,b,c,d,e)] indicate Duncan's pairwise differences among samples from different origins ($P < 0.05$).

lipid metabolism, xenobiotics biodegradation and metabolism, enzyme families, metabolism of terpenoids and polyketides, metabolism of other AAs, glycan biosynthesis and metabolism, and biosynthesis of other secondary metabolites (Table 4).

Genes involved in AA metabolism and carbohydrate metabolism predominated in samples of different origin. Genes involved in AA metabolism, biosynthesis of other secondary metabolites, enzyme families, lipid metabolism, metabolism of cofactors and vitamins, metabolism of other AAs, metabolism of terpenoids and polyketides, nucleotide metabolism, xenobiotics biodegradation and metabolism in the samples from Yunnan were significantly different ($P < 0.001$, Duncan's test) from other origins. Genes involved in carbohydrate metabolism in samples from Sichuan were significantly different from other origins ($P < 0.001$, Duncan's test).

Other functional genes of the bacteria in samples were related to cellular processes, environmental information processing, genetic information processing, human diseases, organismal systems, and others (Table 5). Among these, genes involved in membrane transport were dominant in samples. Genes involved in membrane transport in the samples from Yunnan were significantly different ($P < 0.001$, Duncan's test) from other origins except for samples from Sichuan ($P > 0.05$, Duncan's test).

A heatmap was plotted to estimate the similarities of the samples for the functional genes (Fig. 6). Most of the samples from different origin cluster with each other except for QH1 again. The results were consistent with our previous cluster analysis results of the bacterial community.

## Microbial interactions in samples

In the present study, relationships between genera in the samples were calculated using the Spearman rank correlation coefficient and visualized as a network in Cytoscape

**Table 5 Other functional genes from bacteria in the samples.**

| Function | Percentage composition in samples | | | | |
|---|---|---|---|---|---|
| | Xizang <br> n = 3 | Sichuan <br> n = 3 | Qinghai <br> n = 3 | Gansu <br> n = 3 | Yunnan <br> n = 3 |
| Cell growth and death | 0.58 ± 0.02[c] | 0.44 ± 0.00[e] | 0.57 ± 0.01[c,d] | 0.63 ± 0.00[b] | 0.65 ± 0.00[a] |
| Cell motility | 0.62 ± 0.09[c,d] | 2.28 ± 0.05[a] | 1.54 ± 0.73[b] | 0.58 ± 0.04[c,d,e] | 0.64 ± 0.01[c] |
| Transport and catabolism | 0.17 ± 0.01[c] | 0.20 ± 0.00[a,b] | 0.22 ± 0.04[a] | 0.13 ± 0.01[c,d] | 0.12 ± 0.00[d,e] |
| Membrane transport | 11.34 ± 0.38[d] | 13.51 ± 0.07[a,b] | 12.28 ± 0.63[c] | 11.18 ± 0.04[d,e] | 13.86 ± 0.22[a] |
| Signaling molecules and interaction | 0.34 ± 0.00[a,b] | 0.28 ± 0.00[d] | 0.27 ± 0.05[d,e] | 0.35 ± 0.00[a] | 0.33 ± 0.00[a,b,c] |
| Signal transduction | 1.35 ± 0.04[c,d,e] | 2.28 ± 0.02[a] | 1.67 ± 0.25[b] | 1.36 ± 0.02[c,d] | 1.53 ± 0.01[b,c] |
| Folding, sorting and degradation | 2.40 ± 0.06[b,c] | 2.15 ± 0.00[d] | 2.24 ± 0.12[d,e] | 2.59 ± 0.02[a] | 2.50 ± 0.02[a,b] |
| Replication and repair | 8.20 ± 0.07[c] | 7.44 ± 0.08[e] | 8.14 ± 0.12[c,d] | 8.56 ± 0.04[b] | 11.04 ± 0.18[a] |
| Transcription | 2.34 ± 0.05[d] | 2.63 ± 0.01[a] | 2.55 ± 0.17[a,b,c] | 2.31 ± 0.01[d,e] | 2.55 ± 0.03[a,b] |
| Translation | 5.42 ± 0.07[b,c] | 4.77 ± 0.06[e] | 5.26 ± 0.18[c,d] | 5.55 ± 0.03[b] | 7.15 ± 0.12[a] |
| Cancers | 0.14 ± 0.01[c] | 0.11 ± 0.00[de] | 0.12 ± 0.01[d] | 0.20 ± 0.00[a] | 0.16 ± 0.00[b] |
| Cardiovascular diseases | 0.02 ± 0.00[bc] | 0.01 ± 0.00[d,e] | 0.02 ± 0.01[b] | 0.02 ± 0.00[a] | 0.01 ± 0.00[d] |
| Immune System diseases | 0.06 ± 0.00[e] | 0.08 ± 0.00[b,c] | 0.07 ± 0.00[b,c,d] | 0.08 ± 0.00[b] | 0.10 ± 0.00[a] |
| Infectious diseases | 0.44 ± 0.01[c,d] | 0.55 ± 0.00[a] | 0.46 ± 0.02[b] | 0.41 ± 0.00[e] | 0.44 ± 0.00[c] |
| Metabolic diseases | 0.10 ± 0.00[b] | 0.09 ± 0.00[d] | 0.10 ± 0.00[b,c] | 0.09 ± 0.00[d,e] | 0.11 ± 0.00[a] |
| Neurodegenerative diseases | 0.51 ± 0.05[a,b] | 0.29 ± 0.01[c,d] | 0.34 ± 0.11[c] | 0.61 ± 0.01[a] | 0.28 ± 0.03[c,d,e] |
| Circulatory system | 0.09 ± 0.01[a,b] | 0.04 ± 0.00[c,d,e] | 0.05 ± 0.03[d] | 0.12 ± 0.00[a] | 0.05 ± 0.01[c,d] |
| Digestive system | 0.04 ± 0.00[a] | 0.02 ± 0.00[b,c,d] | 0.03 ± 0.01[b] | 0.03 ± 0.00[b,c] | 0.01 ± 0.00[e] |
| Endocrine system | 0.14 ± 0.02[d] | 0.21 ± 0.00[a,b] | 0.22 ± 0.07[a] | 0.08 ± 0.01[d,e] | 0.10 ± 0.00[c,d] |
| Environmental adaptation | 0.12 ± 0.00[b,c,d] | 0.10 ± 0.00[e] | 0.12 ± 0.00[b,c] | 0.12 ± 0.00[b] | 0.16 ± 0.00[a] |
| Excretory system | 0.00 ± 0.00[d,e] | 0.02 ± 0.00[a,b,c] | 0.02 ± 0.01[a] | 0.02 ± 0.00[a,b] | 0.01 ± 0.00[a,b,c,d] |
| Immune system | 0.06 ± 0.00[a] | 0.04 ± 0.00[c,d] | 0.04 ± 0.02[c] | 0.06 ± 0.00[a,b] | 0.02 ± 0.00[e] |
| Nervous system | 0.05 ± 0.00[e] | 0.10 ± 0.00[a] | 0.06 ± 0.01[b,c] | 0.06 ± 0.00[c,d] | 0.07 ± 0.00[b] |
| Unclassified | 15.43 ± 0.05[a,b] | 15.61 ± 0.00[a] | 14.48 ± 0.71[c,d] | 14.57 ± 0.06[c] | 13.74 ± 0.08[e] |

**Note:**
Letters [(a,b,c,d,e)] indicate Duncan's pairwise differences among samples from different origins ($P < 0.05$).

software (Fig. 7). The network for microbial community in samples consists of 47 nodes and 244 edges. A total of 36 bacterial genera were found to be hub genera ($\geq 6$ edges per node) in the network, such as *Lactococcus, Lactobacillus, Leuconostoc, Acinetobacter, Pseudomonas, Enterococcus, Enterobacter, Enhydrobacter, Bacillus, Chryseobacterium, Staphylococcus, Janthinobacterium, Pseudomonas, Sphingomonas,* and others. The network is dominantly cooperative, and the ratio of cooperative vs. non-cooperative interactions is 115:7. All the non-cooperative interactions were found to be related to the genera *Lactobacillus, Acetobacter,* and *Gluconobacter. Lactobacillus* had a negative relationship with six other genera: *Lactococcus, Enterococcus, Enterobacter, Erwinia, Salmonella,* and *Serratia. Acetobacter* was negatively associated with four other genera: *Pseudomonas, Camobacterium, Sediminibacterium,* and *Jeotgalicoccus. Gluconobacter* was negatively related to four other genera: *Bacillus, Brevibacterium, Sediminibacterium,* and *Tetragenococcus.* Other bacterial genera were positively associated with each other.

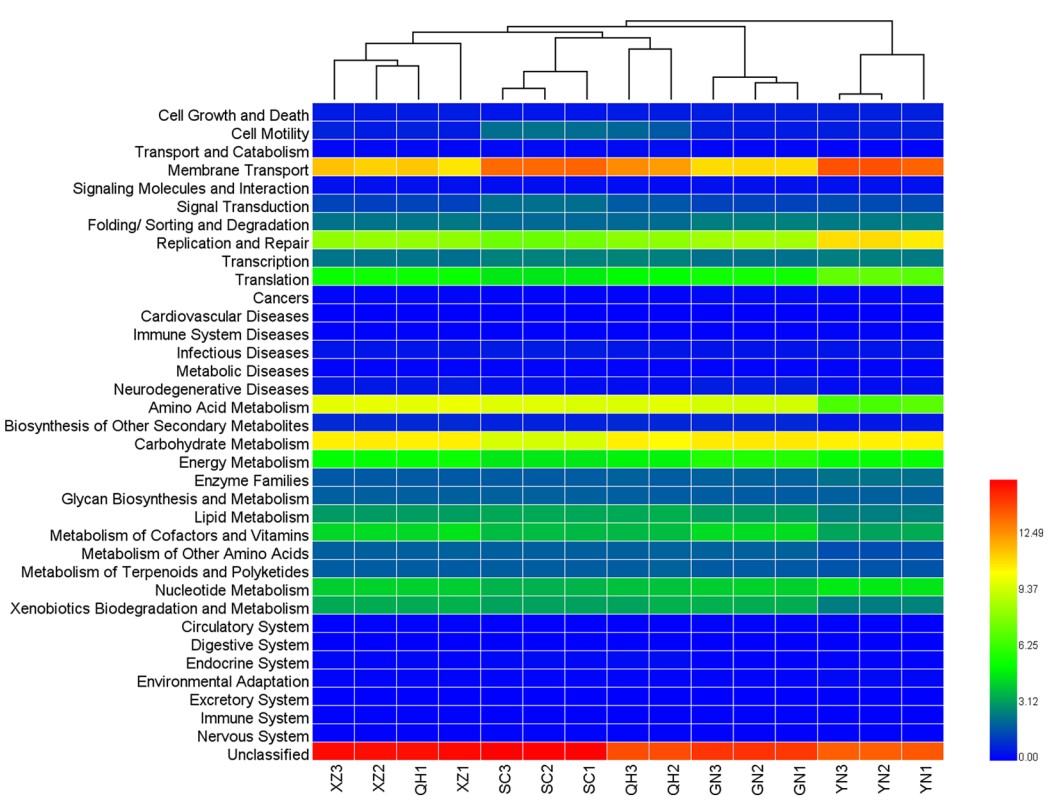

**Figure 6 Heatmap of clustering analysis based on the functional genes in the samples.**

## DISCUSSION

Qula is a special naturally fermented dairy product and has been made in the Qinghai–Tibetan Plateau for centuries (*He et al., 2011*; *Li et al., 2010*). It is considered an analog of cheese and prepared from milk of Yak. As Qula was generally made by natural fermentation without inoculating any commercial starter cultures, it is a typical case study for food biogeography. In this research, bacterial diversity in Qula from Yunnan, Xizang, Qinghai, Gansu, and Sichuan of China was evaluated using high-throughput sequencing of the V3–V4 regions of the 16S rRNA gene. The result showed that microbiota in Qula samples from different origins exhibit a large degree of biodiversity and were influenced significantly by geography. Similar results were found for the fermented milk matsoni and tarag (*Sun et al., 2014*; *Bokulich et al., 2015*). In the production of Qula, no starter is used to inoculate unpasteurized yak milk, and naturally occurring microorganisms serve as the inoculums instead. Therefore, the process of fermentation is largely affected by the local "house flora" (*Santos et al., 1998*). The five sampling locations have different local "house flora," which may lead to significant differences in the microbial community between samples. Similar reports were found for the fermented yak milk in Tibet, Plaisentif cheese in Italy (*Wu et al., 2009*; *Dalmasso et al., 2016*). Milk composition may also influence fermentation microbiota in dairy fermentations (*Sun et al., 2014*; *Bokulich et al., 2015*; *Dalmasso et al., 2016*).

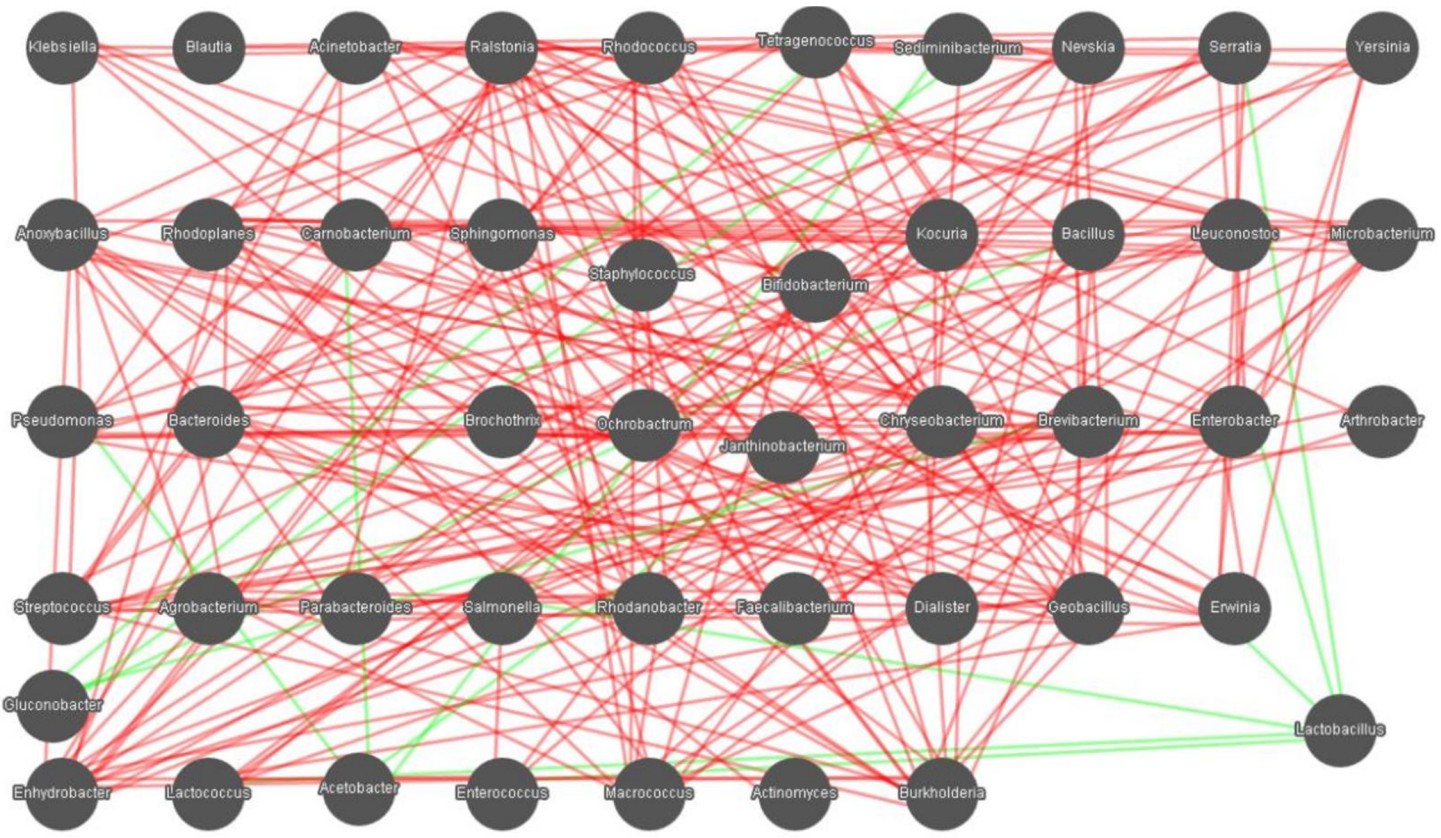

**Figure 7 Networks of microbial interaction in the samples.** Each node represents a bacterial genus, edges denote significant correlations between phylotypes (Spearman's ρ > 0.6). Positive correlations colored in green indicate co-occurrence, whereas negative correlations in red indicate mutual exclusion.

The composition of yak milk is affected by many factors including breed, diet, and parity. Milks from different yak breeds can contain widely different contents of fat, protein, lactose, and other nutrients (*He et al., 2011*). Diet is another important factor influencing the profile of yak milk, which is greatly affected by season (*Liu et al., 2011*). *Liu et al. (2011)* also reported differences in composition of yak milk between primiparous and multiparous animals. More work is needed to assess the effect of milk composition on microbiota in raw yak milk and Qula. In the present study, Qula from Yunnan stored for a shorter time (two weeks) had the lowest bacterial richness, while other samples stored for a longer time (more than one month) had higher bacterial richness. Since Qula is usually packaged in bags and kept for a period of time at room temperature before sale, environmental factor will continue shaping the microbiome of Qula during storage. Further studies are needed to assess what role storage condition and time may play on the profile of Qula microbiota.

A variety of microorganisms including various lactic acid bacteria, coliform bacteria, and aerobic bacteria have been found in Qula collected from Qinghai and Gansu (*Zhang et al., 2015*). In the present study, a total of 6 bacterial phyla and 73 bacterial genera were identified in the Qula samples from five different regions. Among them, Firmicutes,

Proteobacteria, Actinobacteria, and Bacteroidetes were also observed in raw milk (*Zhang et al., 2017*) and fermented milk products such as Plaisentif cheese (*Dalmasso et al., 2016*) and tarag (*Sun et al., 2014*), but Acidobacteria and Cyanobacteria were not detected in previous studies (*Sun et al., 2014*; *Dalmasso et al., 2016*; *Zhang et al., 2017*). Most of the bacterial genera detected in the present study were also observed in tarag, such as *Lactobacillus*, *Streptococcus*, *Lactococcus*, *Leuconostoc*, *Acinetobacter*, *Acetobacter*, *Enterobacter*, *Enhydrobacter*, *Bacillus*, *Macrococcus*, *Chryseobacterium*, *Klebsiella*, *Citrobacter*, and *Bifidobacterium* (*Sun et al., 2014*). Among them, *Lactobacillus*, *Acetobacter*, *Lactococcus*, *Leuconostoc*, and *Streptococcus* were dominant in tarag (*Sun et al., 2014*). Some of the bacterial genera were also observed in raw milk, including *Pseudomonas*, *Staphylococcus*, *Stenotrophomonas*, *Janthinobacterium*, and *Sphingomonas* (*Zhang et al., 2017*). Numerous bacterial genera such as *Lactobacillus*, *Lactococcus*, *Leuconostoc*, *Pseudomonas*, *Staphylococcus*, *Stenotrophomonas*, *Chryseobacterium*, *Enhydrobacter*, *Enterococcus*, and *Sphingomonas* have been observed in Plaisentif cheese (*Dalmasso et al., 2016*). In previous studies, some species of bacterial genera such as *Lactococcus*, *Lactobacillus*, *Enterococcus*, *Bifidobacterium*, and *Weissell* have probiotic potential (*Sun et al., 2010*, *2014*; *Zhang et al., 2017*). However, members of *Staphylococcus*, *Enterobacter*, *Serratia*, and *Burkholderia* have been involved in human infections as opportunistic pathogens (*Iversen & Forsythe, 2003*; *Munsch-Alatossava & Alatossava, 2006*; *Raquel et al., 2016*; *Visscher et al., 2017*; *Zhang et al., 2017*). Members of the genus *Staphylococcus* frequently occur in milking animals; for example, *Staphylococcus aureus*, which is the most prevalent pathogen in dairy ruminants and often detected in milk and milk products (*Visscher et al., 2017*; *Zhang et al., 2017*). Several strains of the genera *Enterobacter*, *Serratia*, and *Burkholderia* are pathogenic and are considered to be associated with raw milk (*Iversen & Forsythe, 2003*; *Munsch-Alatossava & Alatossava, 2006*; *Raquel et al., 2016*). The results, whether or not the traditionally prepared homemade Qula is safe, needs further verification.

PICRUSt is a computational approach that can predict the functional composition of a metagenome from 16S data and a reference genome database (*Langille et al., 2013*). Such integrated analysis is proved to be cost-effective and helpful for combinational evaluation of the functional and taxonomic properties of the microbiota from human, soils, mammalian guts, and others (*Inoue et al., 2017*; *Pavloudi et al., 2017*; *Zeng et al., 2017*; *Wu et al., 2018*). In the present study, AA metabolism and carbohydrate metabolism were the main metabolic pathways identified for bacteria in all Qula samples, similar results were found for the raw milk from goats (*Zhang et al., 2017*). The dominant metabolic pathways may relate to some certain bacteria in certain circumstances (*Zhang et al., 2017*). PICRUSt predictions could also likely be improved by including habitat information in a predictive model, because some genes might correlate strongly with environmental parameters as well as phylogenetic similarity to reference organisms (*Langille et al., 2013*). In the present study, the reason why bacterial communities would shift with respect to functional genes cannot be explained due to lack of specific environmental information. Hence, further studies on isolation and cultivation of bacteria from these samples are needed, in order to illuminate metabolic pathways of certain bacteria in certain circumstances.

Microorganisms do not exist in isolation, but instead form complex and interacting ecological webs (*Lidicker, 1979*; *Konopka, 2009*; *Freilich et al., 2010*). Microbial networks have been inferred for a range of communities, from soil and ocean communities to human body communities (*Faust & Raes, 2012*; *Zhou et al., 2010*; *Arumugam et al., 2011*). In the bacterial network, most of bacterial genera were positively associated with each other. All the negative interactions were related to *Lactobacillus*, *Acetobacter*, and *Gluconobacter*. *Lactobacillus* can produce either alcohol or lactic acid from sugars, which is mainly applied to ferment traditional food and beverages (*Makarova et al., 2006*). *Acetobacter* is characterized by the ability to convert ethanol to acetic acid in the presence of oxygen, and is used for producing vinegar (*Cleenwerck et al., 2002*). *Gluconobacter*, which can also oxidize ethanol into acetic acid, was related to beer spoilage and rot in fruits (*Spitaels et al., 2014*). Their metabolic products (e.g., alcohol, lactic acid, acetic acid) have anti-bacterial activity and can inhibit the growth of other pathogenic and food spoilage bacteria, which was in agreement with some previous studies (*Sun et al., 2014*; *Zhang et al., 2015*). Since compositional data have a negative correlation bias (*Pearson, 1897*), more rigorous approaches including SPARCC (*Friedman & Alm, 2012*) and SPieCeasi (*Kurtz et al., 2015*) should be applied to analyze correlation to illuminate the relationship of the microbes in the process of Qula production and storage.

## CONCLUSIONS

Pyrosequencing has been shown to be a powerful tool in exploring a large diversity of natural environments. However, until recently few studies have considered food microbiota. In this study, microbial diversity and communities in Qula samples from different regions in China were studied using high-throughput sequencing. This is the first study to apply this technology to study food ecology in Qula. The results provided insights into the impact of environment on bacterial communities in Qula and lay a foundation for further investigations into the food ecology of Qula.

### Funding

This work was supported by Special funds for discipline construction of GAU (GAU-XKJS-2018-247), Program for Fu Xi Talents in GAU (FXYC20130110), the National Natural Science Fund of China (31560442, 31760466, 31460425), and the Enterprise research transformation and industrialization project (No. 2018-SF-C29) for financial support. The funders had no role in study design, data collection and analysis, decision to publish, or preparation of the manuscript.

### Grant Disclosures

The following grant information was disclosed by the authors:
GAU: GAU-XKJS-2018-247.
GAU: FXYC20130110.

The National Natural Science Fund of China: 31560442, 31760466, 31460425.
The Enterprise research transformation and industrialization project: 2018-SF-C29.

## Competing Interests

The authors declare that they have no competing interests.

## Author Contributions

- Yan Zhu conceived and designed the experiments, performed the experiments, analyzed the data, contributed reagents/materials/analysis tools, prepared figures and/or tables, authored or reviewed drafts of the paper, approved the final draft.
- Yingying Cao performed the experiments.
- Min Yang authored or reviewed drafts of the paper.
- Pengchen Wen authored or reviewed drafts of the paper.
- Lei Cao analyzed the data, contributed reagents/materials/analysis tools.
- Jiang Ma performed the experiments, analyzed the data, prepared figures and/or tables, sRA submission.
- Zhongmin Zhang performed the experiments.
- Weibing Zhang conceived and designed the experiments, contributed reagents/materials/analysis tools, authored or reviewed drafts of the paper, approved the final draft.

## Data Availability

This dataset is available in the SRA at the NCBI under accession number SRP128600.

## Supplemental Information

Supplemental information for this article can be found online at http://dx.doi.org/10.7717/peerj.6044#supplemental-information.

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
