# Peer review of "Bacterial diversity and community in Qula from the Qinghai–Tibetan Plateau in China"

_PeerJ, doi:10.7717/peerj.6044_

## Round 0.1 · original submission · Major Revisions

Your manuscript has now been reviewed by two researchers with specific expertise on microbial community ecology, microbiome profiling and metagenomics. As such, the reviews are both thorough and well-constructed, and both agree that there are fundamental issues with the manuscript that preclude acceptance in PeerJ.

Both reviewers agreed that the basic profiling of the microbial communities in Qula are valid and a reasonable addition to the literature, but both reviewers emphasized that the conclusions drawn about the safety and geographic variation in Qula microbial communities were unjustified and exceeded the scope of the data presented.

Both reviewers also expressed considerable reservations, which I agree with, on the use and interpretation of PICRUSt as a method of predicting the metagenomic structure of Qula microbial communities.

Both reviewers provided extensive comments and recommendations for improvement; for the manuscript to be accepted you must carefully and thoroughly address every one of their points. I personally agree with many of the constructive criticisms both reviewers made, and in many cases they are in firm agreement. As such, I would expect that your revised manuscript would look very different, both in terms of presentation of findings, figures, and discussion of conclusions, and I would encourage you to incorporate most of the reviewer recommendations or justify your decision to not make the recommended changes.

Reviewer 1 ·

Basic reporting

This manuscript reports phylogenetic and functional diversity of microbial communities in Qula from five geographically distinct regions in China. The authors conclude that microbial phylogenetic and functional diversity varied by geographic region. In addition, the authors suggest that their results indicate that traditionally made Qula may be unsafe to consume following long storage time and/or poor packaging (e.g. see lines 252-253). The authors make a compelling case for the importance of understanding microbial dynamics in Qula. However, there are major issues including:

1) A potentially inaccurate description of the methods (i.e. was this a metagenomics study or not? See Experimental Design #1 & 2).

2) The methodology is not clearly defined and as a result it is unclear what the “triplicate” Qula samples represent (e.g. are samples #1-3 from one/multiple Qula/yak/producers? See Experimental Design #3).

3) There is inadequate background information to contextualize the results (e.g. how and why (be specific) might Qula microbial communities differ as a function of the five distinct geographic locations? See Basic Reporting #1; Experimental Design #3, & Validity of the Findings #1).

4) There is insufficient data reported in the manuscript to support the authors conclusions (e.g. what evidence is there to suggest that poor packaging and long storage times lead to unsafe Qula? See Validity of the Findings #2).

5) Other issues regarding the text and figures/tables (see Basic Reporting #2 & 3), as well as the validity of the findings (see Validity of the Findings #3).

BASIC REPORTING
1. Literature references, sufficient field background/ context provided:
- A major issue with the current study is the lack of background information on the five geographic locations of Qula collection (i.e. Yunnan, Xizang, Qinghai, Gansu, and Sichuan). In the absence of any measured and/or discussed differences based on geographic location, it is unclear why and how Qula microbial communities might vary with location. Simply suggesting that geographic location, and thus environmental factors, may have impacted the microbial communities is too vague to be insightful. A discussion of how “elevation, climate, temperature, oxygen level, and atmospheric pressure” (lines 226-227) varied across the five locations would strengthen this manuscript.
- The literature discussion in the Introduction is too brief and fails to provide a comprehensive summary of previous research. For example, Zhang et al. (2015) used rRNA sequencing to study bacterial communities as a function of Qula development, suggesting that more than just “numbers of microbes” (line 47) was assessed prior to the current study. I believe the authors are trying to distinguish between culture independent (current study), and culture dependent (e.g. Zhang et al. 2015) studies of Qula communities (compare lines 44-47 to 47-48). This is an important distinction that should be stated more clearly to maintain proper representation of previous work.

2. Professional article structure, figs, tables:
The authors have organized the manuscript into professional and coherent sections and have made all raw data available in the SRA at the NCBI.

Figures and Figure Legends:
Fig 1: Ordering the legend to match the stacked rarefaction curves would enhance the readers ability to quickly line-up each curve with geographic location (just a suggestion).
Fig 2: There is no information in the text or figure legend suggesting that we should compare across the bars in the order in which they are connected. As a result, the extra color in the plot is ultimately confusing (e.g. I stared at the plot for a while trying to figure out what light pink indicated). Additionally, converting to a log-scale would enable visualization of the less abundant groups (i.e. currently cannot see Acidobacteria or Cyanobacteria).
Fig 3: See comments for Fig 2.
Fig 4: This is a complicated plot and requires a much more descriptive legend. Further, based on the 4 lines of text that references this figure (lines 170-174) it shows differences in bacterial OTUs between regions – how is this different from Fig 7?
Fig 5: What level of bacterial taxonomy is being compared? Need to include this in the legend.
Fig 6: Why does QH1 cluster with other QH samples in this figure, but with the XZ samples in the following dendrogram (Fig 7)?
Fig 8: This figure would be significantly more informative if there was some level of identification of the KO #’s. For example, organizing these into higher level classifications (e.g. amino acid metabolism) would help the reader to quickly assess what major metabolic differences are present in the geographically distinct communities. Again, QH1 is not clustering with the other QH samples, but with the XZ samples, and this should be addressed.
Fig 9: Though I am not familiar with network plots, Fig 9 is only referenced in lines 204-206 and it is not clear to me how this information informed the conclusions for this study. Could this figure be removed?

Tables and Table Legends:
Table 1: Might the Qula microbial communities differ based on the color of the grain? Yunnan had the lowest diversity and based on this table it seems that color of grain, rather than geographic location, may have influenced the Qula communities. How long have these Qula’s been stored (e.g. line 252)? How “badly” was the Qula packaged (e.g. line 253)? Do #1-3 (e.g. YN1 – YN3) represent triplicate samples collected from a single Qula purchased at each location? Or are samples #1-3 three different Qula’s bought from the same (different?) market(s) at each location? This table, as well as the text, require significantly more detail to understand the sampling methods employed.

3. Clear and unambiguous, professional English used throughout:
I commend the authors for their clear and professional writing. However, there are multiple minor spacing and grammatical errors, as well as ambiguous portions of the text that need to be addressed – I have highlighted some of these here:

Line 20 – “and a total of 47 OTUs at different levels significantly differed…” what does this comparison mean if they are at different levels?
Line 21 – “The functional genes of the bacteria present in samples also indicated differences in bacterial communities between the samples…” this is misleading and needs to clearly state that functional genes were not directly assessed, but rather predicted based on phylogeny
Line 44, 44, 45, 53, 64, 77 (need to check throughout paper) – multiple locations throughout the text require a space to be inserted.
Minor grammatical errors including line 76 (‘wereas” should be were as), line 89 (“sequence” should be plural), line 112 (what does “respectively” refer to?), and line 131 (“bacterial and bacterial phylotypes”).
Line 49 – “;” should be removed
Line 63 – It is unclear what “five types” of Qula refers to. Are “types” equivalent to geographic location? If so, are samples #1-3 representative of triplicate samples from a single Qula? Or three different Qula’s purchased from each location? Other? This needs to be clearly stated in the methods.
Line 140-145 – This paragraph is confusing as it is unclear which samples exhibited significantly different species richness, and which did not. Additionally, it is not clear to me what “also” in line 144 refers to – geography, and what else?
Line 186 – Is “douchi” meant to be “Qula”?
Line 209 – “Obviously, the network…” The distinction between dominantly cooperative and non-cooperative interactions is not obvious to me. A brief introduction to network analysis would strengthen this section by providing the context required to assess these plots.
Line 227 – “Sun et al. 2010” should not be red
The first paragraph in the Discussion section suggests that Qula may be unsafe to consume because of the presence of pathogenic bacteria; however, the second paragraph suggests that the presence of “alcohol, lactic acid, acetic acid” producing bacteria may inhibit the growth of pathogenic bacteria. These contradictory speculations are never addressed simultaneously and as a result, the reader is left unsure whether Qula is unsafe or safe to consume. A more comprehensive and cohesive discussion of the study would strengthen this manuscript.

Experimental design

The research question - to assess “diversity of microbial communities in Qula from different regions in China” (line 12) – is well defined. In addition, this study contributes to a growing understanding of food ecology of Qula (line 274) through the use of culture-independent molecular techniques (e.g. line 55). However, there are major issues with the experimental design:

1. A crucial distinction that needs to be addressed in future iterations of this manuscript is whether a metagenomics analysis was, or was not, conducted. Line 65 states that Qula samples were collected for “metagenomics DNA [analysis]”; however, the following sections suggest that primers were used to amplify 16S rRNA genes and that PICRUSt was used to infer metabolic function from phylogeny. My understanding is that isolating and sequencing 16S genes is not a metagenomics analysis. The use of “metagenomics” needs to be addressed throughout the manuscript.

2. In the absence of a metagenomics analysis, a compelling case needs to be made for using phylogeny of Qula microbial communities to infer community metabolic function.

3. The sampling method is not clearly defined. For example, it is not clear whether the 3 samples (#1-3) for each of the 5 geographic locations represent 3 different Qula made by different producers (and possibly 3 different Yak), 3 Qula produced from one producer (maybe one Yak), or whether the 3 samples are triplicate samples from one Qula. Without a clear understanding of the samples collected, it is difficult to assess whether the differences in microbial communities resulted from different source animals, producers of the Qula, environmental factors that may vary by geographic location, etc.

Validity of the findings

1. As stated in Basic Reporting #1 & Experimental Design #3, there is insufficient background information on the Qula samples to assess whether the microbial communities differed as a function of geography, grain type (i.e. white vs yellow), producer of the Qula, age of the Qula, packaging of the Qula, etc. If there is compelling evidence to suggest that the differences between samples are due to geography, a detailed discussion of why and how geography may alter Qula communities is needed.

2. I have found no evidence in this manuscript to support the authors conclusion that “badly-packed Qula” and “long storage times” may increase the colonization of Qula by pathogenic bacteria. If data on these two storage methods was collected, it needs to be included.

3. There is no discussion of the clustering of QH1 with the samples from XZ, for both the community and functional analyses. A discussion of why this sample might be less like other QH samples, and more like samples from XZ, is needed.

Reviewer 2 ·

Basic reporting

The writing needs substantial work for clarity and grammer. Please see details in attached review.

Experimental design

The presented objective does not match the analyses. The research objective is presented as an analysis of regional effects, but no metadata are included that would enable analysis of regional drivers. Substantial revision of analyses and additional data are needed if the paper is to be presented as a regional analysis. Otherwise, I think the data are still publishable but the paper needs to be revised to present a simpler goal (a basic description of the bacterial communities of Qula).

Statistical methods need some clarification.

Please see attached review.

Validity of the findings

Validity of findings depends on issues described above (writing, objectives, and statistics). In addition, the PICRUSt and network analyses need more thoughtful treatment. Please see attached review.

Additional comments

Please see attached review.

Annotated reviews are not available for download in order to protect the identity of reviewers who chose to remain anonymous.

---

## Round 0.2 · Major Revisions

We appreciate the revised manuscript, but must ask that you do further editing to your Response to Reviewers document before we reconsider the manuscript, as this document is currently not acceptable.

In your response to the reviewers, there are many statements where you simply state something to the effect of "this was fixed in the revised manuscript". This places an undue burden on the reviewers and editor. The burden of revision and review is on the authors. Please extensively revise your Response to Reviewers as follows:

For each query, explain HOW you revised the manuscript to consider the recommendation. Ideally, you will quote the original and revised sections of the manuscript, to highlight the changes you made. In cases where the changes are very extensive/long, point the reviewers to specific line numbers in the revised manuscript encompassing the revised section.

Avoid simply deleting sentences that a reviewer questions...this is not a solution except in cases where reviewers specifically indicate that a sentence should be removed. Instead, revise or clarify the statement.

---

## Round 0.3 · Major Revisions

Reviewer 2 noted that you did not address any of their major comments, possibly because you felt that only those comments that highlight specific lines in the manuscript should be addressed. The reviewer's major comments are the most important, and require a detailed and thorough response and amendment. Please review both reviewers recommendations on both drafts of your manuscript, address R1's minor additions and R2's original major points, and return a revised draft. Thank you.

Reviewer 1 ·

Basic reporting

1. Literature references, sufficient field background/ context provided:
Reference to “elevation, climate, temperature, oxygen level, and atmospheric pressure” was removed and the authors provide a compelling discussion of the potential for yak breed, diet, and parity to impact differences in Qula microbial communities. In addition, they discuss the potential of house flaura, and storage time, to influence Qula microbial communities in the absence of starter inoculum.

- The authors have sufficiently addressed my original concern that the literature discussion in the Introduction was too brief and failed to provide a comprehensive summary of previous research.

2. Professional article structure, figs, tables:
Fig 1: This figure is not re-ordered in my copy of the manuscript...but again, this is not a huge deal.

Fig 2 & 3 (now Tables): were converted to tables and though these are easier to read than the original figures, I find it much easier to view changes in community composition in a stacked bar plot. The authors may consider including stacked bar plots of community composition in addition to the tables.

Fig 4 (now Fig 7): The authors updated the figure legend and the figure is now easier to follow.

Fig 5 (now Fig 3): The authors updated the figure legend to include the level of taxonomy.

Fig 6 (now Fig 4): QH1 has been highlighted throughout the results as not clustering with the other QH samples. However, “.family” needs to be updated.

Fig 8 (now Fig 6): The authors have updated this figure and it is now much easier to decipher broad changes in potential community function.

Tables and Table Legends:
- Lines [55-56] indicate the Qula samples were purchased directly from different sellers at local markets in five regions. I recommend explicitly stating that samples #1-3 are “not representative of triplicate samples from a single Qula” to make this completely unambiguous to the reader.

3. Clear and unambiguous, professional English used throughout:
- There are still grammatical errors and missing spaces in my copy of the manuscript.

- The authors provided a clear explanation of the Qula samples in their response to my review, however, I recommend explicitly stating under the Sample Collection section that "samples #1-3 are not representative of triplicate samples from a single Qula", but rather non-replicated samples "bought from different producers at each location".

- [Line 206] I very much appreciated the authors explanation of Fig 7 in their comments to my review; however, this is not addressed in the text and if it is not obvious to all readers, I would suggest updating the text and/or removing “obviously”.

- I commend the authors for updating the text throughout the manuscript to reflect the limitations of their study and the requirement for further experimentation to understand whether Qula is safe to consume.

Experimental design

The authors have done a nice job making clear that this was not a metagenomic study (it might be helpful to explicitly state this was 16S amplicon sequencing in the abstract). However, I would still recommend including a discussion of other studies/ motivation for using PiCRUST to assess metabolic potential of the communities.

Validity of the findings

The authors have added a well written and cited discussion of how microbial communities on Qula may be impacted by yak breed, diet, and parity, as well as house floura and storage times. In addition, the authors have updated their discussion to reflect the limitations of their study.

Additional comments

In case it's easier to read through, I have included a word document with my updated review in blue.

Annotated reviews are not available for download in order to protect the identity of reviewers who chose to remain anonymous.

Reviewer 2 ·

Basic reporting

Unable to evaluate. Revisions did not address my major concerns from the previous review.

Experimental design

Unable to evaluate. Revisions did not address my major concerns from the previous review.

Validity of the findings

Unable to evaluate. Revisions did not address my major concerns from the previous review.

Additional comments

The authors have not addressed any of my major concerns, therefore I cannot review the manuscript in its current form. My review was structured in two parts: The first part of the review described major problems with the paper, where I devoted a paragraph to each of the following: (1) overview, (2) Basic characterization versus regional analysis, (3) statistics, (4) issues with PICRUSt, (5) issues with network analysis, (6) writing. These are major, general issues not associated with any specific line number in the manuscript. The authors have not addressed any of the major points.

The second part of my review listed manuscript lines to which I had miscellaneous or specific questions. The authors have responded to these, but given that the major revisions needed are not addressed at all it does not make sense for me to review these minor revisions.

---

## Round 0.4 · Minor Revisions

Your paper has been re-reviewed and one reviewer has indicated that there are still a few minor points that require clarification. I agree with the reviewer that it is important to the manuscript that these be carefully addressed, and that the review be read and responded to carefully, with changes documented in the text of the manuscript. As the reviewer indicates, please avoid responding only to the reviewer: any information you give the reviewer is likely to improve the paper as well and I prefer that you directly change the manuscript and copy those changes into the rebuttal where needed.

Reviewer 2 ·

Basic reporting

Please see general comments.

Experimental design

Please see general comments.

Validity of the findings

Please see general comments.

Additional comments

The authors have made substantial revisions to the manuscript, particularly the discussion, description of statistical analyses, and quality of the writing in general. I appreciate their hard work to improve the paper. Many of my concerns have been completely addressed. A few areas still need a little work. These are straightforward and I am confident that the authors can rectify them with minor additional revisions. I have outlined these below, with the history of communications regarding each point pasted in for context. My most recent comment in each case is annotated "Reviewer Response".
Note to the authors: please address all issues in the manuscript itself, not just in your response to me. Thank you.

Issue 1:
Initial Reviewer comment: What environmental factors differ between regions? How are they likely to affect the Qula microbiome?

Author Response:Thank you for your advice. The previous discussion was not proper and they were revised in the revised manuscript. In the production of Qula, no starter is used to inoculate unpasteurized yak milk, and naturally occurring microorganisms serve as the inoculums instead. Therefore, the process of fermentation is largely affected by the local ‘house flora’. Different environmental factors lead to different ‘house flora’. The five sampling locations have different local ‘house flora’, which may lead to significant differences in the microbial community between samples. These were included in the discussion (see lines 234-238 in the revised manuscript).

Reviewer response: This is interesting. Please provide a reference for the statement that different regions have different “house flora”.

Issue 2:
Initial Reviewer Comment: The study uses PICRUSt to predict the Qula functional genome. With PICRUSt, the usefulness of the result depends on whether the PICRUSt database includes genome sequences of isolates that are closely related to the bacteria in the sample being studied. I recommend demonstrating the validity of the PICRUSt data in two ways. First, state if there is there a threshold (minimum percent identity between the sample 16S read and database hit) for a genome to be included in analyses. Second, report the proportion of the bacterial community in the sample is represented by PICRUSt data. (If the community is mostly made up of taxa that are rare in PICRUSt’s sequenced genomes, then most of the data acquired from PICRUSt will link to taxa that are relatively rare in the study sample, and therefore the overall PICRUSt data will not represent the community functional genetic profile.)

Author Response: Thank you for your advice. We use a ‘closed-reference’ OTU picking protocol, and the threshold of Picking OTUs for use in PICRUSt is 97% (see lines 106-107 in the revised manuscript). OTUs that a KEGG profile could not be retrieved constitute the fraction of unexplained taxonomic units (FTU) (Aßhauer et al., 2015), they were included in the tabe S1 (see lines 108-110 in the revised manuscript).

Reviewer Response: The manuscript states “The threshold of Picking OTUs for use in 
PICRUSt is 97%.” (L. 106.) Please clarify: does this mean that every OTU in your study had 97% similarity with a sequenced genome in the PICRUSt database? (Other than the few that were in the FTU fraction, which was only about 10% of OTUs for most samples.)

Issue 3:
Initial Reviewer Comment: L 259 [line number from first draft] “In the bacterial network, most of bacterial genera were positively associated with each other.” I think the authors should discuss the possibility that this result is an artifact of relative abundance data. For example, if one very common taxon decreases in relative abundance, then several other taxa may increase in relative abundance simply because total relative abundance has to add up to 1. This problem is not unique to the current study – it is a problem with all relative abundance data – but because this study has little additional information to help determine the biological plausibility of the interactions it’s particularly important to acknowledge the possible artifact due to working with relative abundances.

Author Response: Thank you for your advice. Yes, the limitation is existed in the present bacterial network. In the further studies, the bacterial genera in Qula samples in different fermentation time or storage time will be used for network analysis.

Reviewer Response: I think the authors have misunderstood my comment. For a more detailed explanation, please see papers such as Gloor et al 2017 “Microbiome Datasets Are Compositional: And This Is Not Optional” in Frontiers in Microbiology). The important point is that you need to state, in the paper, the possibility that correlations among relative abundance in your data may be an artifact of compositional data, and may not mean anything regarding the ecology of the microbial communities at all.

Issue 4:
Initial Reviewer Comment: Throughout the paper, bacterial taxa are distinguished as being “dominant” or not dominant, but there is no definition of what it means to be dominant. How is the dominance status of an OTU determined? For examples, see L. 155-169. L. 193-198 gives a similar example with respect to predicted functional genes.

Author Response: Thank you for your advice. The dominant taxa has a relative abundance >1%.

Reviewer Response: This issue is still confusing in the manuscript. It does not seem like dominance is only based on a relative abundance of >1%. For example the abstract reads: “A total of six phyla were identified with relative abundance >1% in the samples, and Firmicutes and Proteobacteria were dominant.”

Issue 5:
Writing and grammar: The authors have improved the writing. However, there are still some typos and grammatical errors. E.g. line 34 (“dispersion of fresh yak”, should probably read “dispersion of fresh yak milk”), line 39 “molecularbiology” is not one word. These are only two examples. Please assess the whole manuscript once again for errors.

---

## Round 0.5 · Minor Revisions

Thank you in advance for addressing the few minor revisions recommended by the reviewer. Once these are addressed we expect we will be able to accept your manuscript.

Reviewer 1 ·

Basic reporting

Please see general comments.

Experimental design

Please see general comments.

Validity of the findings

Please see general comments.

Additional comments

The authors have put a lot of time into revising this manuscript. Though I did not receive a response from the reviewers for this last round of reviews, I have read through the manuscript and see that most of my suggestions have been adequately addressed. I am confident that with a few minor revisions they can address the remaining issues.

Writing and grammar:
There are still numerous lines throughout the manuscript that are missing spaces. E.g. line 76 and 80 “(…Caporaso, et al., 2012).Sequence…”; “…(Edgar, et al., 2011).Quality…”

Line 99: I believe “was” is meant to be “were” (“Differentially abundant features among the different samples was identified using…”

Line 108: I believe this sentence is missing a few words “OUT’s that a KEGG profile could not be retrieved constitute the fraction…” This could read ‘OTUs for which a KEGG profile…’

Line 114: I believe this sentence is meant to read “Spearman’s rank correlations between selected genera were calculated using the (insert name) R package…”

This is not a comprehensive list and I would encourage the authors to do a thorough review of the entire manuscript for errors such as these.

Citations
Are the citations in line 98 correct? They don’t refer to the adonis (PERMANOVA) or ANOSIM packages in R, and a quick ctrl-F doesn’t bring up either acronym in either paper.

Lines 279-281 are a bit misleading as, for example, Pavloudi et al. 2017 do not use PICRUST to predict microbial functional diversity, they use the R package Tax4Fun. I bring this up because it appears there may be differences in the quality of functional predictions based on environment (e.g. Iwai et al. 2015; DOI: 10.1371/journal.pone.0166104).

Reviewer 2 ·

Basic reporting

Please see General Comments.

Experimental design

Please see General Comments.

Validity of the findings

Please see General Comments.

Additional comments

The authors have addressed all of my critiques. I thank them for their continued efforts revising this manuscript. I have no further concerns.

---

## Round 0.6 · accepted · Accept

Thank you for the hard work on the manuscript!

#